# Evolving Narratives in Tourism and Climate Change Research: Trends, Gaps, and Future Directions

Kaitano Dube [1,2]

1   Kaitano Dube Ecotourism Management, Faculty of Human Sciences, Andries Potgieter BlvD, Vanderbijlpark 1911, South Africa; dubekaitano@gmail.com; Tel.: +27-710-096-290
2   Faculty of PhD in Aviation Management, Emirates Aviation University, Dubai Academic City, Dubai, United Arab Emirates

**Abstract:** This study presents a comprehensive overview of the evolving landscape of tourism and climate change research over the past decade by leveraging bibliometric analysis and a dataset sourced from the Scopus Database. The study scrutinised over 3400 English language articles. The analysis reveals a remarkable surge in publications, signifying the growing recognition of climate change's multifaceted impact on tourism. However, a noteworthy geographical disparity emerges, with many regions remaining underrepresented in the literature, particularly in Africa and the Middle East. This oversight is concerning, given the vulnerability of these regions to climate change and their burgeoning tourism industries. The study also highlights the pivotal role of influential scholars, funding organisations, and publication outlets in shaping the research landscape. The European Commission and the National Natural Science Foundation of China are major funders. At the same time, journals like *Sustainability* and the *Journal of Sustainable Tourism* serve as prominent platforms for disseminating research findings. The analysis uncovers thematic trends, including the growing focus on climate change modelling and its implications for destination planning. However, research gaps persist, notably in sports tourism and climate resilience within the tourism sector. In conclusion, this study offers valuable insights into the current state of tourism and climate change research, pinpointing areas that demand increased attention and inclusivity. It is a valuable resource for scholars, policymakers, and stakeholders working towards a sustainable and resilient future for the global tourism industry in the face of climate change.

**Keywords:** extreme weather events; carbon emissions; climate risk; tourism destinations; vulnerability

## 1. Introduction

The climate change debate remains one of today's most topical issues [1,2]. As much as climate change has been considered a geographic and environmental issue, today, the debate has been extended to all facets of life. The climate change debate is closely intertwined with the global developmental agenda. As such, climate change has been central to adopting and localising Agenda 20230 for Sustainable Development as encompassed in the Sustainable Development Goals (SDGs). Various economic sectors have exhibited multiple levels of vulnerability to the impacts of climate change [3]. Consequently, various economic sectors have sought to address climate change within their context.

One such sector is the tourism sector. The tourism sector has a long history of paying attention to climate and weather [4], given their central role in shaping tourism activities. Many tourism activities rely heavily on ideal climatic and weather conditions. Consequently, the sector has developed various tourism climate indexes [5,6] to understand how climate change can disrupt various activities as the climate continues to alter globally. Ordinarily, one would expect that climate change debates and coverage in studies have evolved with time. The interest in understanding the relationship between tourism and climate change dates back numerous years ago [7], with climate change and tourism having a dual relationship [8,9].

Simply put, tourism affects climate change in several ways, including producing carbon emissions [10,11] throughout its value chain, which causes global warming, which is a critical driver of climate change. On the other hand, tourism activities depend on ideal climatic parameters for certain activities. Consequently, alterations of climatic parameters at various destinations can positively or negatively affect tourism activities [10]. Consequently, various tourism climate change indexes have been developed for various tourism activities [6,11] to better understand the role of climate in each tourism-specific activity.

It is believed that focus and attention have been evolving as the understanding of climate change has evolved across space and time [12–14]. The United Nations board is tasked with tackling climate change, the United Nations Framework Convention on Climate Change through the Intergovernmental Panel on Climate Change's annual Conference of Parties (COP), which shapes the global agenda surrounding the climate change agenda and debates. It is equally expected that the focus and attention on tourism studies and climate change will revolve around speaking to the needs and focus at that particular time and for various geographic regions.

Tracking these developments in the industry can sometimes be a challenge in the absence of synthesis studies. This study highlights critical debates over recent decades, which have been revolutionary in the climate change debate in many respects. The period also coincides with one of the worst pandemics ever to affect the world in recent memory, the COVID-19 pandemic [15]. Given the increase in the intensity of extreme weather events [16,17], the cost of climate change and the occurrence of COVID-19 remains a matter of international interest. The advent of COVID-19 has seen a reinvigoration of greater demand for environmental protection and sustainability [18] within the tourism sector [19–21]. These sector developments could have a long-lasting and transforming impact on the tourism industry.

This study, therefore, seeks to track evolving narratives in tourism and climate change research to track trends, gaps, and future directions. The study is critical for the theoretical development of tourism and climate change and can play a significant role in informing sector response strategy. To achieve this, several theoretical underpinnings informed the study of conceptualisation thinking and design. The vulnerability and resilience theories of sustainable development are at the top of these lists. The tourism vulnerability theory is applicable as it allows us to understand tourism destinations' key risks and vulnerabilities from a climate change perspective and how the sector responds to build a resilient and sustainable tourism future. The approach has been used in previous climate change tourism studies by Calgaro et al. [22]. The approach allowed the researchers to examine the complex climate change and tourism interplay, ranging from climate change impact to mitigation, adaptation, and resilience building. Ensuring the sector's sustainability was covered by making critical borrowings from the sustainable development theory. The theory assists in the interrogation of tourism from a sustainable development perspective. The climate change debate has its roots in sustainability [23]. The 2030 Sustainable Development Goals agenda is squarely aimed at ensuring that the world achieves its developmental agenda while addressing the challenges imposed by climate change.

## 2. Research Methodology

The study uses a bibliometric analysis research approach protocol. According to Donthu et al. [24] bibliometric analysis is a popular methodological approach that allows for the analysis of large volumes of data and allows the researcher to track the field or a particular discipline to draw development and insights. It allows the researcher to identify emerging themes and gaps. It allows for a better understanding of the development of a particular field or domain [25]. This forms the basis of the importance of this study on climate change in tourism. Bibliometric analysis is growing in popularity across various fields [26,27].

The Scopus Database was used in this regard. The Scopus Database was used because it is one of the largest databases comprising carefully selected books, conferences, and

journal articles. It is a trusted database with an international reach. The data archived and indexed by the Scopus database come from various fields, and they are compatible with other research bibliometric analysis tools such as VOSviewer and Cite Space. They are widely accepted for use in various scientific fields [28,29].

Over seven thousand publishers use the database, with over 1.8 billion cited references dating back to 1970. Within the database, there are more than 84 million records. The material on Scopus is rigorously vetted and selected by an independent review board of experts in their fields [30]. This makes the information reliable for academic and policy planning purposes. The embedded analytical tools allow users to search and filter information. The key search terms used for this study were "tourism AND climate AND change". The search criteria included the article title, abstract, and keywords. The search yielded 3556 articles. Of these articles, 3400 articles were written in English. These are the only articles that were considered for this study. The remaining articles were excluded due to a lack of capacity for interpretation.

The initial process was to run the Scopus analyser, which provided a simple analysis of the articles. A basic analysis revealed top authors, contributing institutions, documents per year, and the institutions. Some data were exported, and graphs were produced using Microsoft 365 Analysis Toolpak. For detailed analysis, articles were exported as CSV files. The CSV files were later uploaded to VOSviewer version 1.6.19 for analysis. A co-occurrence analysis for all keywords using the full counting method was conducted. The threshold for a maximum number of occurring words of keywords was set at 5. Out of the 15,418 words, 1504 met the threshold. For each of the 1504 keywords, the total of the co-occurrence links with other keywords was calculated, reducing the number of keywords to 1000.

R Studio version 4.3.1 (16 June 2023 uctr) was used for additional analysis using the same CSV file. All the figures that were produced used the above-stated applications. VOSviewer has often been used for bibliometric analysis, including tourism studies. It has strong analytical capabilities, which assisted a great deal in this research.

## 3. Results and Discussion

The study found that there has been significant growth in tourism climate change literature between 2012 and 2022 (see Figure 1). Regardless of the two downward fluctuations witnessed in 2015 and 2017, the number of publications has increased, with publications almost tripling over the study period. This indicates the significant importance of such studies and the acknowledgement of the threats posed by climate change to the sector.

The study also found that in as much as there has been a notable growth in the number of publications conducted thus far, several areas have not been covered in the current studies. This point to knowledge gaps in some geographic regions (Figure 2). The unavailability of tourism and climate change data in several African countries is worrying, given that there is an understanding that Africa is one of the most vulnerable regions to climate change. This puts the tourism product at risk. In the absence of studies focussed on Africa's largely nature-based tourism product, it remains unknown how climate change affects tourism, and policy planners and implementers cannot act in a manner that builds resilience and ensures sustainable development.

The United States of America (USA) dominates the number of studies on tourism and climate change conducted over the period in question. It is followed by the United Kingdom, Australia, China, Spain, Canada, Germany, Italy, and South Africa, topping the countries with the most publications. Therefore, the tourism climate change narrative seems skewed toward the global north, dominating the research space.

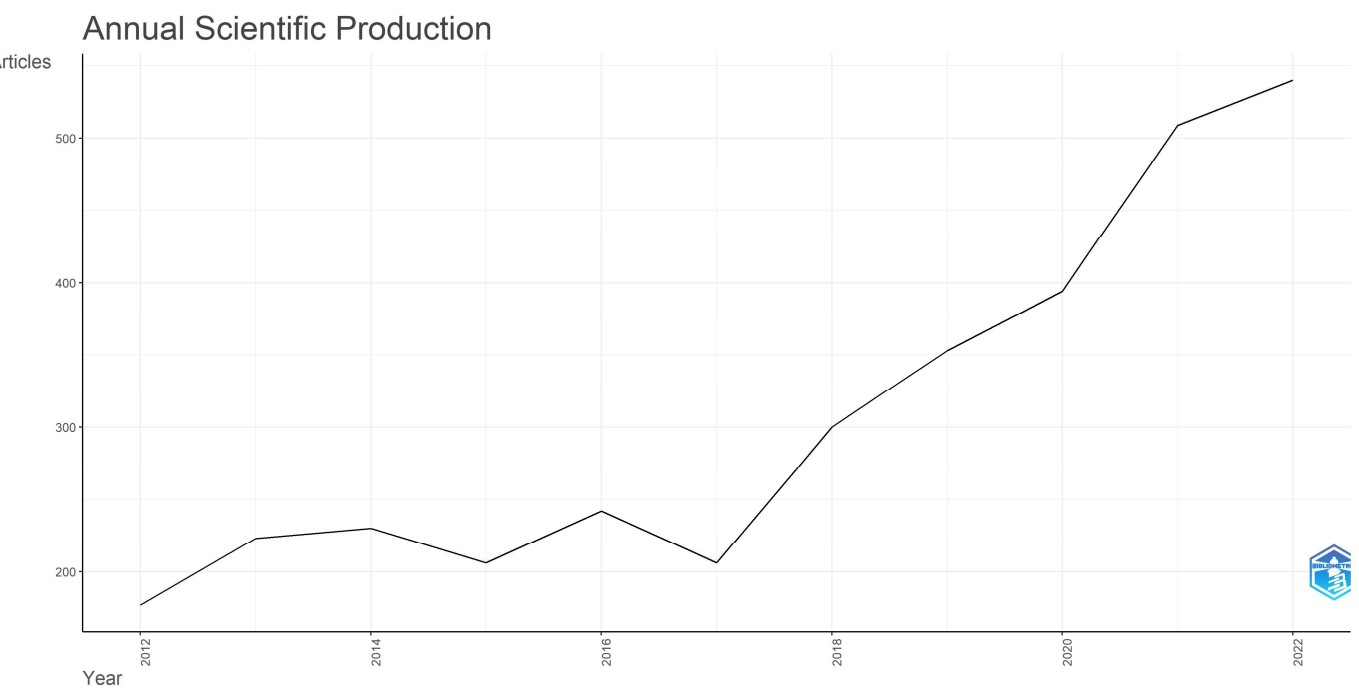

**Figure 1.** Annual publication tourism and climate change.

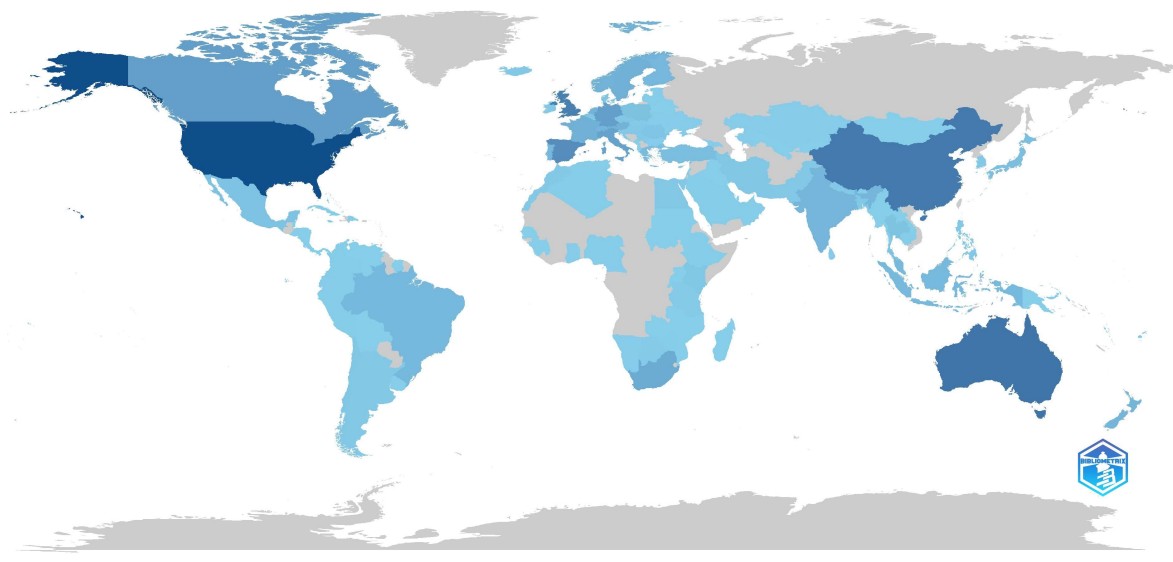

**Figure 2.** Global distribution of Scopus indexed tourism climate change data.

The affiliate universities of the authors have benefited from this research work, with the University of Waterloo (Figure 3a) leading research on tourism and climate change. This shows the contribution of Daniel Scott, who leads globally in terms of tourism climate change research contribution (Figure 3b). Several Chinese Academy of Science authors have impacted the tourism climate change research space. The tourism climate change space has a couple of authors who have dominated the research space covering various aspects of tourism and climate change space. These researchers who have made a global impact are Scott, Daniel, Gössling, Stefan and Hall, and Colin Michael. The individual and collaborative work of these three individuals has significantly shaped the tourism and climate change debate.

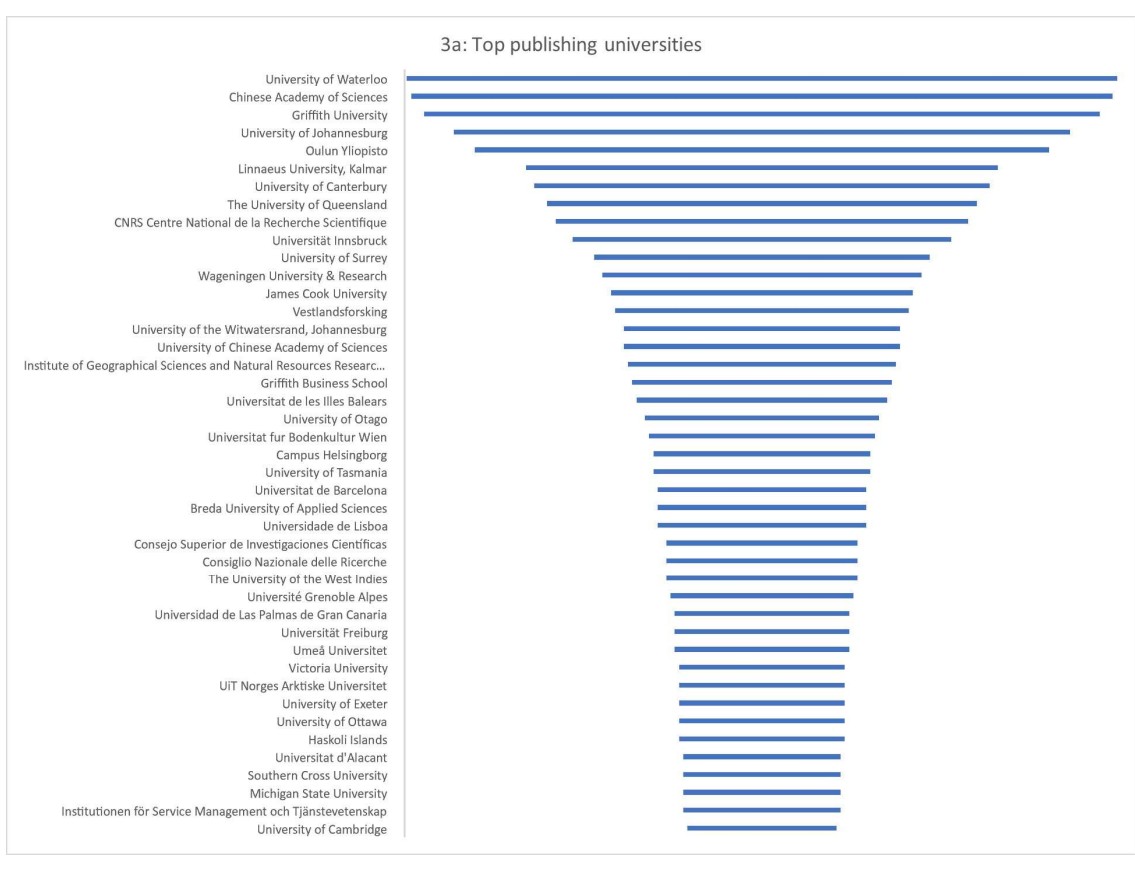

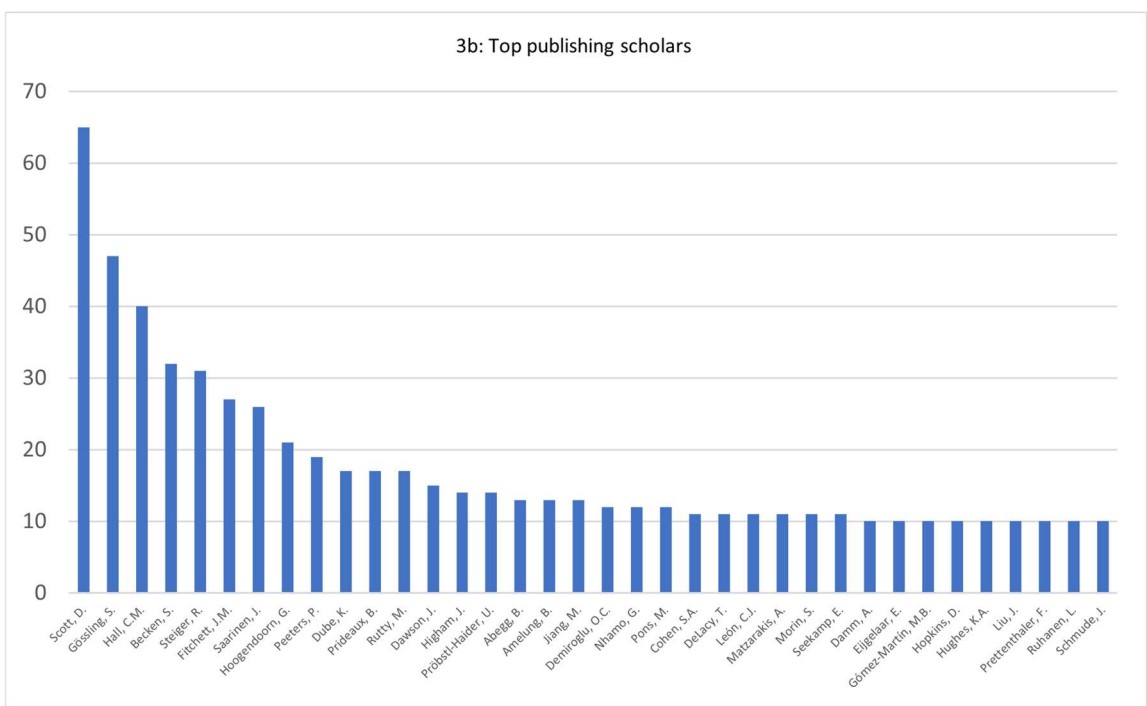

**Figure 3.** Some of the top scholars in tourism and climate change research.

Other scholars who have made some contributions in this space include the contributions of Susanne Becken, an Australian research scholar (Griffith University) who is probably one of the prominent women working in the tourism climate change research space. Robert Steiger's research has largely focused on climate change and ski tourism [31–33]. Steiger significantly contributed to the tourism and climate change research space. Another

woman significantly contributing to the tourism climate change nexus is Jennifer Fitchett, a South African tourism geographer (Witwatersrand University, Johannesburg) who has written extensively on the tourism climate change index [34–36] and tourism's vulnerability and adaptation [37,38]. Her collaboration with another tourism geographer, Gijsbert Hoogendoorn (University of Johannesburg) [39,40], in this space, greatly impacted the tourism and climate change debate from a Southern African perspective. Jarkko Saarinen has worked with tourism geographers from the global north and south to put issues of tourism climate change in the spotlight and has emerged as one of the top scholars in this space. His collaborative work with scholars from South Africa and Botswana [41–43] has shed light on the contentious issue of tourism and climate change (Figure 4).

**Figure 4.** Collaboration network in tourism and climate change: Source Author.

Paul Peeters worked with several other tourism scholars and is one of the top three networked scholars researching various aspects of tourism and climate change. Part of the work covered during the period in question includes a review of tourism and water use [30], aviation and climate policy [31], and tourism and decarbonisation. The research

focus, which has been focused on tourism, aviation, and climate change, primarily focuses on climate change decarbonisation [44].

One of the remarkable developments is the emergence of a sizeable number of scholars from the global South. In addition, scholars from the University of Johannesburg and the University of Witwatersrand University, Johannesburg Dube Kaitano, Vaal University of Technology and Godwell Nhamo, the University of South Africa, have made a significant contribution to examining the climate change nature tourism nexus. Most of the studies focussed on responding to the knowledge gaps on iconic tourism destinations in Southern Africa, such as the hydrological challenges caused by droughts at Victoria Falls waterfalls [45,46], the Okavango Delta [47], Kruger National Park [15], and Cape Town, [48] amongst others, are critical tourism assets in the region.

Figure 4 shows that a few clusters of authors dominate the tourism climate change research space. There is also an overconcentration of research in some regions. There is a cluster led by the top scholars, forming a small nuclear that dominates the debate. The authors follow a pattern of the well-to-do economically developed countries where the bulk of research is also concentrated. There is very little cross-collaboration between research clusters, with most research work being conducted in rather isolated clusters. There is a need to ensure that the writing space is democratised to allow for a multiplicity of voices in that critical research space. Improved collaboration between clusters is also critical to harmonise and provide a more comprehensive field picture. This has to happen between research clusters representing various geographic regions.

### 3.1. The State of Funding for Tourism and Climate Change Research

Several funding organisations have come to the fore to fund tourism and climate change research, and there is a definite close link between funding and research output. Several studies have been funded by the National Natural Science Foundation of China (Figure 5). The funding has helped to show that research from the Chinese Academy of Science has become the second highest in tourism climate change research outputs.

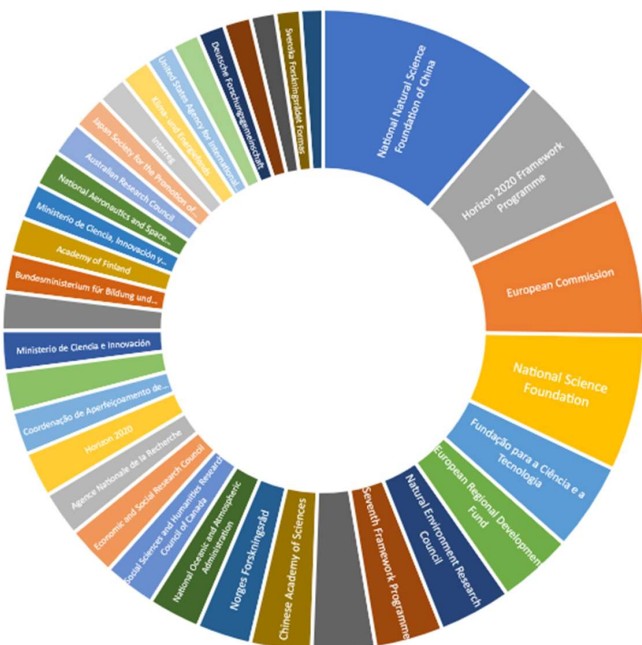

**Figure 5.** Funding agencies for tourism and climate change research.

The European Commission and the Horizon 2020 Framework Programme are the other significant tourism climate change research funders. This funding has been central in propelling European academics to succeed, as evidenced by the number and volume of published research in the territories. The research output from that region rewards the

National Science Foundation of America's investment in tourism climate change. The areas with less research output, such as Africa, do not seem to have many funders for tourism and climate change research. The lack of funding in this respect is worrying as it hampers the quality and quantity of research in the area. While a few collaborators from the global north have worked with African scholars, this has not been adequate to cover all the regional gaps, given the geographic knowledge gaps reported earlier.

Publishers publish their work on tourism and climate change in several journal outlets. The clear top outlets for this kind of research are *Sustainability* and the *Journal of Sustainable Tourism*, which are the clear leaders in this regard (Figure 6). Several leading tourism geographers and several top authors have publications in these journals. Despite the relentless attack and criticism on platforms like Trinet, *Sustainability* remains a favourite for tourism and climate change publishers. While there is no evidence of some top authors having published in it, the IOP Conference Series 'Earth And Environmental Science' is the third most populous publication outlet. It is equally an open-access publication. The availability of tourism and climate change in open access allows consumers to interact with some of the latest research findings on tourism and climate change.

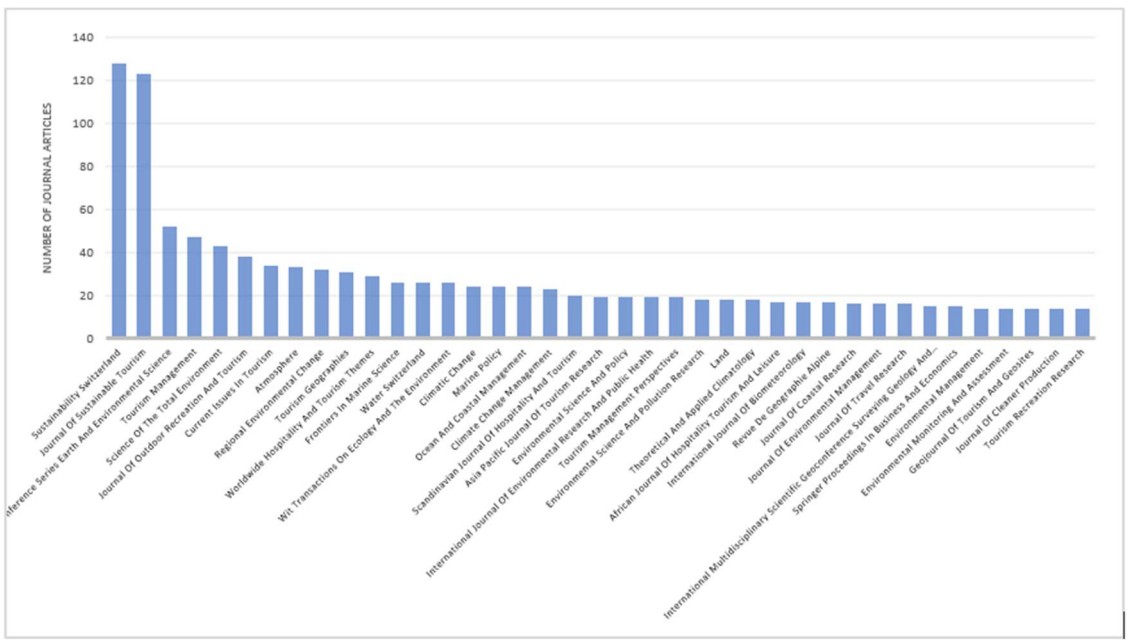

**Figure 6.** Leading journal outlets for tourism and climate change.

The wide range of journal outlets reflects the transdisciplinary nature of tourism and climate change research conducted during that period. This scope is expected to continue diversifying and growing as the understanding of climate change evolves. This also talks to a multiplicity of topical issues covered under the tourism climate change space, which is an area that is growing to include other branches of human society intertwined in the tourism space.

### 3.2. Thematic Trends in the Tourism and Climate Change Research Space

In leading tourism, geographers have focused their research over the past years on covering various destinations with to the primary aim of understanding climate impacts on tourism development [49–52]. This central trend dominated the study between 2015 and 2020 (Figure 7). Within this framework, various types of tourist destinations have been covered, such as examining the effects of climate change on various countries considered tourist destinations. Countries such as South Africa, Peru, New Zealand, Australia, China, and the United Kingdom, amongst others, are some of the areas which dominate the studied destinations. Concerning South Africa, Jong et al. [50] argued that carbon emissions posed

a risk to the country's tourism industry. Jong et al. [50] then called for action to address carbon pollution to ensure the sustainability of the tourism industry. The sustainability of the tourism industry in South Africa was found to be continually compromised by the threat of climate change and extreme weather events such as droughts [53], global warming [52], and sea level rise [49,51], amongst others. Saarinen et al. [54] (called for robust action to ensure that the tourism sector adapts to the many threats of climate change in South Africa.

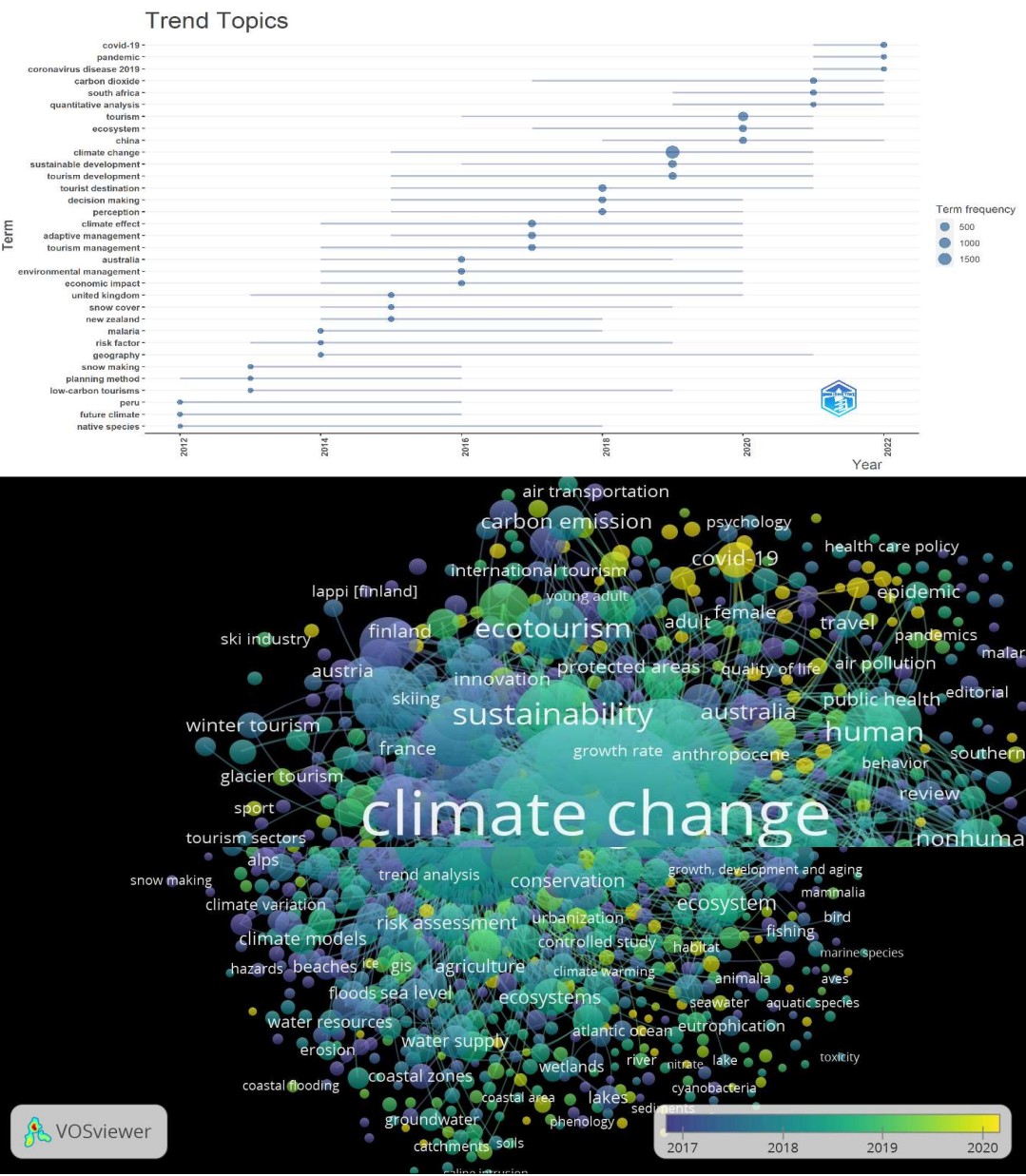

**Figure 7.** Themes and trends on tourism climate change research between 2012 and 2022.

Figure 7 highlights areas of research concentration over the period of study. Evidence shows that climate change is the most researched topic, with climate change being a central theme dominating research space. Around 2012, climate change was primarily perceived as a future event [55] as some areas perceived it then. In addition, few studies focussed on native species that form the core of nature tourism being threatened by climate change. For example, Hughes and Convey [56] raised issues with the interface between climate change and native species in Antarctica. Concerns were also raised by Ballantyne and

Pickering [57] about ecotourism's compounding risk of climate change on native species, amongst others.

There has been a consciousness of the need for tourism to be geared towards low carbon as a climate change mitigation mechanism. This was a central theme in 2013; several authors addressed this issue [58–61]. The need to address tourism's carbon footprint by various stakeholders also became a central issue during the discussions on tourism degrowth during and after the Paris Agreement and the adoption of the agenda in 2015. It has generated about 500 articles between 2016 and 2022. It was indeed one of the central thematic issues to be addressed by tourism post-COVID-19 as part of the tourism degrowth. Issues of debate ranged from raising concerns over carbon emissions and the need and strategies to deal with the carbon burden [62–64]. Others like Li et al. [65] raised concerns over the increasing carbon burden and growth of carbon emissions from tourism activities.

There has been a fair attempt to look at tourism stakeholder's perceptions of the impacts of climate change on tourism. These studies took various focus forms, with some looking at how tourists perceive such impacts on the sector [66]. In contrast, others focussed on tourists and destination response to the same, while others looked at tourists' perceptions of mitigation measures adopted by the tourism industry [67–70]. At least 500 articles were published between 2015 and 2020. Such studies are crucial given that tourism is a perception-driven industry. In tourism, perception is reality.

The COVID-19 pandemic centralised the discussion in the tourism climate change space between 2021 and 2022, which became a topical issue, with more than 1000 articles published. While the issues of climate change and COVID-19 might seem remote at face value, a lot was said about how COVID-19 affected the carbon emissions from the tourism industry in both the hospitality and travel industry [71–73].

Some studies also sought to examine the sector's resilience from climate change and the COVID-19 pandemic, while other studies focussed on obtaining insights into the sector's future under the two challenges of COVID-19 and climate change [74–79]. Over the same period, interest has grown in understanding the human health impact of climate change and COVID-19, with the focus also extended to mental health [80]. Chandran et al. [81] focussed on examining the impact of changing climate on the spreading of zoonic diseases, given that the pandemic was a zoonic disease. Before that, the issue of malaria and changing geographic spread was a major topical issue between 2014 and 2018. Du and Ng [82] noted that vector-borne diseases such as malaria and dengue fever were increasing in range due to climate change, which posed a challenge to the tourism industry. Ninphanomchai et al. [83] examined the relationship between meteorological events and the spread of vector-borne diseases such as malaria and their nexus with tourism. This is a critical debate as diseases can raise the cost of visiting destinations and shape travel patterns and intentions.

### 3.3. Tourism Climate Change and Destination Vulnerability

Some country destinations have dominated the tourism climate change space during the study period. Countries such as Peru (2012–2016), New Zealand (2014–2018), Australia (2014–2019) [84,85], China (2015–2021), South Africa (2019–2022), and the United Kingdom (2013–2020) which has been at the forefront of being researched during the period of study. Some topical issues in some of these countries include the contentious drought and rising sea level in South Africa [86]. Concerning Australia, the Great Barrier Reef and Queensland have been major foci of research in that country, with 81 occurrences and a link strength of 794 and 287, respectively. Coral reefs face multiple threats from global warming and subsequent sea surface temperature increases [87]. Several articles focussing on the impact [88,89], adaptation [90], mitigation [89], and resilience of this World Heritage Site have been produced during the period of study from a tourism and climate change perspective.

Post the UN's Agenda 2030 for Sustainable Development, there has been a growing focus on trying to marry climate change to the SDGs with to the aim of understanding how

climate change impact and response from a mitigation and adaptation perspective would affect the progress and achievement of the 17 SDGs and the 169 targets. Academics have been trying to evaluate how challenges of climate change will affect the progress towards SDGs from a tourism perspective.

There is evidence of concerns being raised over some destinations, particularly coastal destinations, which are prone to coastal flooding and seawater intrusion, with several studies conducted to examine the effects of climate change on beach tourism and water supply in coastal areas [91,92]. El-Masry et al. [93] and Agulles et al. [94] raised concerns over the impact of rising sea levels on Mediterranean tourism destinations. On the other hand, Psarra et al. [95] argued that the flooding in the Lake District area threatened tourism UNESCO heritage sites. Chikodzi et al. [96] and Henderson [97] also raised concerns about flooding at heritage sites in coastal areas. The studies also point to island tourism destinations as some of the most vulnerable to coastal inundation, flooding, and erosion, which threaten the attractiveness of the destinations [98–100]. The sandy coastline is particularly cited as one of the most vulnerable to rising sea levels and erosion [101].

Considerable attention has also been paid to the vulnerability of Alps tourism destinations, focusing on the impacts on the ski industry [87,88], glacier tourism [89,90], and winter sports [102–104].

### 3.4. Tourism Climate Change and Water Wores

One of the major focus areas of tourism climate change studies has been water from various facets (Figure 8). There were 1000 water occurrences. Water is a central issue, given its multiple uses in the tourism industry. In some instances, water features act as tourist attractions and destinations. Climate change triggers increased water use and demand in the tourism industry [73,105]. Still, climate change increases conflict and demand for water, among other factors, making it complex. Increased competition for water situations created by climate change poses a significant threat to some tourist destinations [94,106]. Climate change also worsens the water pollution challenges for coastal and terrestrial water. Water's chemical and physical components have been seen to be altered by climate change, which requires the tourism sector to be proactive in water usage. There is a need to adopt a sustainable model for water use in the water-intensive tourism industry to ensure the sector's sustainability. Lessons can be drawn from destinations threatened by water shortage, such as Cape Town, South Africa.

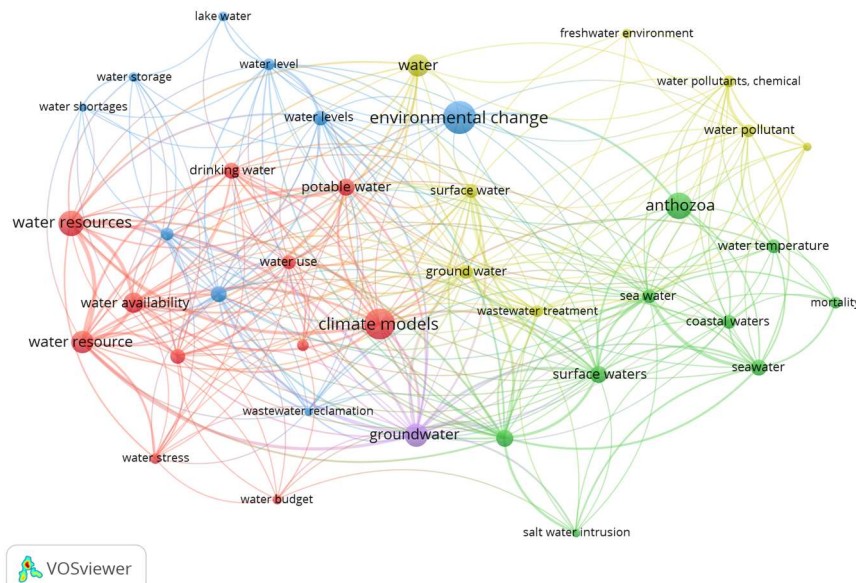

**Figure 8.** Tourism climate change coverage between 2012 and 2022.

*3.5. Research Approaches in Tourism and Climate Change*

Most tourism and climate change studies have used quantitative research techniques (Figure 7). Several studies followed the route of conducting trend analysis. This could include trend analysis of meteorological data as a parameter to gauge changes to climate at various destinations across the world. Time series analysis has 17 occurrences, 100 links, and a total strength of 146. Time series is critical in determining climatic trends, allowing for change detection for various climate parameters.

Geographic Information Systems, a geographic analysis tool, has investigated tourism and climate change studies with 70 occurrences and 489 links. Such analytical techniques are critical in mapping risk areas and overlaying various data layers to gain insight into the changes that might have occurred, giving tourism planners and stakeholders a richer understanding of the issues at play. GIS was used for the spatial analysis of destinations, and in some instances, it was used in processing remote-sensed images. Evidence shows that satellite images had 25 occurrences and 321 total link strength, while spatial analysis had 25 occurrences and 241. Some studies used Land Sat images, which occurred 12 times with a total link strength of 137, which signifies works by tourism geographers who normally use such approaches.

Tourism climate change researchers have also been able to employ and use climate change models to examine the effects of climate change on tourism destinations. Climate change models had 65 occurrences, 310 links, and a total strength of 825. This means that studies have also been able to project the future scenarios of some of the destinations. Using climate change models is critical in allowing tourism stakeholders to plan for the future of destinations under climate change. As such, the usage of climate change models is a welcome development which needs to be extended to other destinations where climate modelling has not been done.

It is interesting to see the usage of controlled studies in tourism and climate change space. The approach has 37 occurrences, 318 links and a total strength of 777. Control studies were used to evaluate climate change impacts in protected areas and other ecotourism destinations to examine the effects of climate change on flora and fauna. Protected areas have been a major focus of attention for tourism climate change studies during the period of study, which occurred 75 times, with a link strength of 387 and a total link strength of 748. Protected areas face existential threats as nature-based tourism destinations from multiple threats of climate change, human encounters, and encroachment as direct and indirect consequences of climate change. Marine and terrestrial biodiversity in protected areas is threatened by extreme weather events and climate variability worsened by climate change [107,108].

## 4. Conclusions and Recommendations

The tourism climate change debate is a growing area of academic engagement with clear evidence of thematic evolution over time. However, despite increasing publications in other geographic spaces, many regions are still starved of tourism climate change research studies. This is particularly true for many areas in North Africa and the Middle East with glaring research gaps. Such glaring gaps are concerning, and warrants research attention. There are many countries in the north and central parts of Africa and the Arab world where tourism's importance is growing. Understanding how climate change will affect desert tourism and coastal tourism features in countries such as the United Arab Emirates is an issue of concern. As a growing global tourism giant, how extreme weather events such as heatwaves affect tourism activities and infrastructure is critical. North Africa and parts of the Middle East have unique tourism products ranging from cultural heritage products to desert tourism experiences. Despite desert tourism's growing popularity and significance in revenue generated, this has not generated sufficient research, even from a climate change experience. Understanding the vulnerabilities will assist in tailor-making resilience and adaptation measures for those areas.

Several studies have been conducted on climate change and winter sports activities in polar regions. However, the full impact of climate change on other sports outside of that region still needs to be better understood. Sports tourism and climate change are emerging topics with the potential for ground breaking research examining various aspects of sports tourism, such as golf, cricket, athletics, car racing, soccer, marathons and other mega sports events, given the huge tourism benefits associated with such sports tourism activities. The growing concern over increasing temperature due to global warming will also adversely affect sports tourism and other special events central to the hospitality and travel subsectors of the tourism industry. There are vast knowledge gaps within this space that must be addressed.

Apart from the information stated above, other areas require attention in tourism and climate change studies. Issues of tourism climate change resilience have been poorly covered over the period in question. In light of the increasing climate change events in the mould of slow and rapid onset events such as fires, which were equally poorly covered during the period under question despite significant losses in many tourism hotspots, there is a greater demand for the tourism sector to build resilience. It is important to note that tourism and climate change resilience remain under-researched areas which require attention to ensure that the sector remains a viable option for addressing various challenges facing the world and ensuring sustainable community development.

There is also a glaring gap in covering tourism from a disaster risk perspective; such knowledge space remains largely unexplored despite the tourism practitioners requiring it to deal with increasing tourism disasters. In this light, there is a need to address tourism and climate change from a disaster risk and response perspective, given the vulnerability of destinations, events and infrastructure, which make part of the tourism product. This area of research and development has implications for several destinations in light of the increased extreme weather events. From a practical perspective, tourism enterprises need to have disaster management plans to deal with the increasing number of climatic threats from extreme weather events. Emerging from that is the need for the tourism sector to start documenting the loss and damage from tourism, as this is a new and evolving field that is central to the sustainability of the tourism sector. The documentation of loss and damage from the tourism sector will also assist in identifying vulnerabilities and areas of focus for resilience building.

Over and above, the increased attention to climate change in tourism underscores the need for the tourism sector to reduce its carbon footprint and keep pushing towards climate neutrality to save the climate-sensitive sector.

**Funding:** This research received no external funding.

**Institutional Review Board Statement:** Not applicable.

**Informed Consent Statement:** Not applicable.

**Data Availability Statement:** Data sharing not applicable. No new data were created or analyzed in this study. Data sharing is not applicable to this article.

**Conflicts of Interest:** The authors declare no conflicts of interest.

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
