# Peer review of "Evolving Narratives in Tourism and Climate Change Research: Trends, Gaps, and Future Directions"

_atmosphere, doi:10.3390/atmos15040455_

Round 1

Reviewer 1 Report

Comments and Suggestions for Authors

The article's topic had the potential to evolve into an interesting one. However, there is a substantial lack of a theoretical background along with a justification for the importance of the study to tourism. Therefore, I suggest the author provide a solid theoretical discussion that can add to the findings. 

Comments on the Quality of English Language

Minor editing is required. 

Author Response

Dear Sir/Madam

Thank you very much for taking time to review the manuscript. We appreciate the effort and have invested time to address all the issues you raised which we belive has enhanced the quality of the submission.

Kind Regards 

Reviewer 2 Report

Comments and Suggestions for Authors

TITLE: “Evolving Narratives in Tourism and Climate Change Research: Trends, Gaps, and Future Directions”

-        This paper correspond for scope of journal. +

-        The title corresponds to the content of the paper.  +

-        This paper analyzed quantity of published research article with topics of  climate change during ten years long period, and its effect on tourism estimation. Among over 35000 papers, in this article analyzed about 3400 papers on English language which represent studies of climate change which clustered according to subject of investigation, geographic places and countries, funding  organizations, boards and associations which supports research about climate change, as well as scientists and Universities and their achievements in qualitative and quantitative study of tourisms related to  climate change.  a few thousand papers, in papers study connected to  estimation differences of climate change  for the development of tourism and contribution of tourism to economic growth in some region.

-        The main question of paper  addressed to study of contribution of scientific investigation and publishing of results about climate change on the development of tourism on the world and how the determined different model of climate change can use for projection  tourism destination in the future under climate change. The study climate change can  contribute to development different type of tourism  (sport tourism, eco tourism, marine tourism, desert tourism etc.) in the future.

-        The aim of research  is not clearly and fully pointed out  on the end of chapter of Introduction

-        Should be pointed out aim of investigation at the end of Chapter of introduction.

-        Key words are appropriate. +

-        Scientific methodology is applied correctly for this type of study. +

-        Results are clearly presented and discussed.

-        Tables, figures, pictures are clear.+

-        Conclusions are partly derived on the basis of research results. In the conclusions are not allowed quote other authors opinions or results. You should delete sentence from line 376 to 378 which contain reference [98–100]

-        This study represents complementary to the previous ones. +

-        Manuscript is acceptable after minor corrections.

Author Response

Dear Sir/Madam

Thank you very much for taking time to review the manuscript. We appreciate the effort and have invested time to address all the issues you raised, which we believe has enhanced the quality of the submission.

Kind Regards 

Reviewer 3 Report

Comments and Suggestions for Authors

While this is a very interesting paper, the author should revise some issues. In the introduction the author should expand the relationships between climate change and tourism. This will inform the current situation about the topics that create more interest and also the topics which are not represented. In this sense, the author should expand the research gap and the research contribution. In the method, the author should provide details about the bibliometric analysis research approach protocol and support the use of the Scopus database with previous references. This will improve the replicability and validity of the research. Also, in the conclusions, the author should expand the theoretical and practical implications, and the opportunities for future research based on the results.

Author Response

Dear Sir/Madam

Thank you very much for taking the time to review the manuscript. We appreciate the effort and have invested time to address all the issues you raised, which we believe has enhanced the quality of the submission.

Kind Regards 

Round 2

Reviewer 1 Report

Comments and Suggestions for Authors

My decision  is to recommend the article for publication. The paper is interesting and will draw attention after publication. Well done to the authors. 

Reviewer 3 Report

Comments and Suggestions for Authors

While the authors have revised the paper, they should expand the implications.